# Future Prospects of Luminescent Silicon Nanowires Biosensors

**DOI:** 10.3390/bios12111052

**Published:** 2022-11-21

**Authors:** Maria Josè Lo Faro, Antonio Alessio Leonardi, Francesco Priolo, Barbara Fazio, Alessia Irrera

**Affiliations:** 1Department of Physics and Astronomy, University of Catania, Via Santa Sofia 64, 95123 Catania, Italy; 2CNR-IMM UoS Catania, Via Santa Sofia 64, 95123 Catania, Italy; 3URT LAB SENS, Beyond Nano—CNR, c/o Department of Chemical, Biological, Pharmaceutical and Environmental Sciences, University of Messina, Viale Ferdinando Stagno d’Alcontres 5, 98166 Messina, Italy

**Keywords:** silicon nanowire, metal assisted chemical etching, luminescence, optical biosensors, proteins, genome, extracellular vesicles

## Abstract

In this paper, we exploit the perspective of luminescent Si nanowires (NWs) in the growing field of commercial biosensing nanodevices for the selective recognition of proteins and pathogen genomes. We fabricated quantum confined fractal arrays of Si NWs with room temperature emission at 700 nm obtained by thin-film, metal-assisted, chemical etching with high production output at low cost. The fascinating optical features arising from multiple scattering and weak localization of light promote the use of Si NWs as optical biosensing platforms with high sensitivity and selectivity. In this work, label-free Si NW optical sensors are surface modified for the selective detection of C-reactive protein through antigen–gene interaction. In this case, we report the lowest limit of detection (LOD) of 1.6 fM, fostering the flexibility of different dynamic ranges for detection either in saliva or for serum analyses. By varying the NW surface functionalization with the specific antigen, the luminescence quenching of NW biosensors is used to measure the hepatitis B-virus pathogen genome without PCR-amplification, with an LOD of about 20 copies in real samples or blood matrix. The promising results show that NW optical biosensors can detect and isolate extracellular vesicles (EV) marked with CD81 protein with unprecedented sensitivity (LOD 2 × 10^5^ sEV/mL), thus enabling their measurement even in a small amount of blastocoel fluid.

## 1. Introduction

Biosensing devices are of great importance for a plethora of applications for health monitoring. Miniaturized transducers with peculiar physicochemical and optoelectronic properties meet the interest of the scientific community, medical facilities and final users for health monitoring and disease prevention [1]. Compared to other materials, Si availability and performance offer a plethora of industrial technologies for modern device realization. Indeed, being easily available at low fabrication cost, silicon is the leading material for the electronic industry and commercial devices. Moreover, it has the great advantage of not being toxic, unlike other materials, such as indium phosphide or cadmium selenide, which are also expensive and toxic, thus posing a threat to patient health. Among different transduction methods, luminescent sensors are some of the standout approaches due to their easy data interpretation, affordability, and real-time analyses. Optical nanosensors are becoming a leading technology with improved sensing performances and smart applications [2]. In fact, Si nanostructures can merge the bulk readiness to novel physical phenomena exhibited by nanomaterials [3], offering a higher surface-to-volume ratio and improved physical performances [4].

However, silicon is an indirect bandgap semiconductor and its application as a light emitter for photonics is quite restricted. To overcome this problem, the relevance in quantum confined silicon nanomaterials has recently expanded [5]. Unlike nanocrystals (NCs) or porous (pSi) Si materials, one-dimensional confined nanostructures are electrically addressed in a planar or vertical architecture design. Indeed, 1D Nanowires (NWs) offer a more robust luminescence and response to mechanical stress over time. In this framework, silicon nanowires are considered one of the most outstanding building blocks for emerging nanodevices, finding flexible applications in electronics [6,7], energy [8,9,10,11], photonics [12,13], and sensing [14,15,16]. However, gaining an efficient room-temperature light emission from Si NWs would represent a great industrial advancement, opening the way to a wide range of new and unexpected silicon photonic applications. Despite the advantages, achieving the industrial requirements is still expensive and ambitious for Si NW fabrication. In general, NW syntheses adopt the vapor-liquid-solid (VLS) approaches [17,18,19] or top-down methods based on lithographies [8,20] that can hardly produce wires with suitable size below the Bohr radius threshold (4.3 nm for Si) to exhibit room temperature light emission from quantum confinement [21,22]. Moreover, the metal catalyst of most VLS approaches is incorporated into the NWs, thereby introducing deep non-radiative energy bandgap trap states, with detrimental results for both electronic and optical purposes [19]. Furthermore, in situ VLS doping is non-uniform and the dopant segregation towards the wire sidewalls also leads to uncontrolled doping levels [23]. Common luminescent NW sensors are in general based on non-emitting Si NWs used as passive substrates and coupled with luminescent probes, while more interesting approaches employ NWs as quenchers in stem-loop configuration and have recently been used for direct luminescence sensing. In this framework, Si NWs with improved compatibility with Si standard technology and flat architecture fabrication are promising solutions for low-cost, scalable, and non-toxic biosensing devices. Indeed, our Si NWs couple their unique physical and optical properties of room temperature luminescence arising from quantum confinement with the high surface-to-volume ratio of about 10^3^ combined with an easy commercial transfer.

In this paper, we propose a low-cost method for the growth of Si nanowires with efficient light emission at room temperature obtained from quantum confinement. The innovative optical luminescence of such Si nanowires fosters their applications in the biomedical field as a sensing platform for the detection of marker proteins, pathogen genomes, and more exotic systems such as exosome or extracellular vesicles.

## 2. Materials and Methods

Fractal Si Nanowires are obtained by metal-assisted chemical etching (MACE) of Si wafer (Siegert Wafer GmbH, Aachen, Germany) using 2 nm of Au (Kurt J. Lesker Company, Jefferson Hills, PA, USA, 99,99% pellet) layer deposited by electron beam evaporation (Kenosystec KM500, Binasco, Italy) and then etched in an HF/H_2_O_2_ solution (solvents from Sigma Aldrich, Merck, Darmstadt, Germany).

Electron Microscopies: Si NWs were analyzed using a scanning electron microscope (SEM) from Supra Gemini field-emission gun from ZEISS (Oberkochen, Germany) and a transmission electron microscope (TEM) from JEOL (Jeol, Paris, France).

Photoluminescence (PL) and Raman measurements were obtained by using a MicroRaman setup HR800 (HORIBA, Palaiseau, France) equipped with several excitation wavelengths and a visible CCD detector.

Functionalization reagents: Streptavidin (SA) and Phosphate Buffer Saline tablets (PBS, 0.01 M phosphate buffer, 25 °C) were obtained from Sigma Aldrich, biotinylated anti-CD81 monoclonal antibody was purchased from LifeSpan BioSciences (Seattle, WA, USA), and commercial small extracellular vesicles were acquired from Hansa BioMed Life Sciences Ltd. (Tallinn, Estonia).

The NW surface functionalizations for proteins detection were performed at room temperature, with 16 h immersion in an SA solution followed by 4 h incubation in the biotinylated antibody solution (anti-CD81 or anti-CRP for specific binding). Each step was followed by 3 rinses in PBS to remove the unbound molecules in excess and the final sensor was washed 3 times with high performance liquid cromatography (HPLC) grade water to rinse off the salt buffer crystal.

sEVs-functionalization: 10 μg/mL of SA + 50 μg/mL of anti-CD81 Ab.

CRP-functionalization: 10 μg/mL of SA + 50 μg/mL of anti-CRP Ab.

HVB-functionalization: the NW surface was modified by (3-glycidyloxypropyl)trimethoxysilane (GOPS) with exposure to 10 mL of GOPS at 120 °C for 4 h in a vacuum chamber ay 200 mbar. After salinization, the samples were immersed for 4 h in the aqueous HBV complimentary probe at 20 μM concentration in a phosphate buffer solution (150 mM at pH 9.2). For HBV hybridization, 100 μL of HBV solution were applied to the reaction area of the NW sensor, followed by 4 min denaturation at 90 °C with the HBV genome. Then, the hybridization reaction was performed by heating at 50 ± 0.1 °C for 2 h, with 3-times rinsing in molecular biology reagent-grade water followed by nitrogen drying.

Photoluminescence measurements: All sensing measurements were performed by incubating the biofunctionalized NW sensors in the derived target solutions at different concentrations for 4 h at room temperature. Afterward, the exposed sensors were washed in PBS to remove the unreacted species, then rinsed with HPLC grade water and nitrogen dried. All photoluminescence (PL) measurements were performed by focusing the 364 nm or line of an Ar^+^ laser onto the sample through a 60× UV Olympus objective at a laser power of about 80 μW measured on the sample. PL spectra were acquired by an HR800 spectrometer (Horiba Jobin Yvon) coupled to a cooled CCD detector. For each concentration, PL measurements were averaged over 5 different points of the same device and evaluated on 3 sample replicates of different Si NW platforms. The calibration curves have been obtained by repeating the PL measurements several times on different sensors. The relative errors over the replicates are lower than 5%.

## 3. Results

### 3.1. Quantum Confined Si Nanowires for Photonic Applications

Our group optimized an original method for Si NWs fabrication from the standard metal-assisted chemical etching [24,25] by using percolative and fractal Au thin film. This low-cost and maskless technique could be integrated with Si technology in addition to granting good control over the morphological, structural, and physical performances of the NWs [26,27]. The realization of fractal arrays of Si NWs is based on subsequent steps, all of which were realized at room-temperature conditions. Initially, the oxide removal procedure is performed by UV-ozone treatment for 5 min and HF-etching of the Si wafer for 5 min. A 2 nm Au film is then deposited by electron beam evaporation at room temperature onto the clean wafer. Figure 1a displays the scanning electron microscope (SEM) image of a 2 nm Au percolative thin film obtained by electron beam deposition in high vacuum conditions. As sketched in Figure 1b,c, the synthesis continues with the chemical wet etching of Si wafer in a H_2_O_2_ and HF solution, catalyzed by the thin discontinuous layer of metal (Figure 1c). The local oxidation of the Si wafer occurs below the metal covered areas, forming a thin SiO_2_ layer at the metal and Si interface, which is then precisely etched. Finally, after the etching, the gold film is removed with a specific gold etchant KI solution as confirmed by structural analyses. This procedure avoids the detrimental incorporation of the metal catalyst along and inside the wires. Another improvement of this NW synthesis is provided by the fine control on the properties of the nanomaterial. In fact, the H_2_O_2_/HF ratio and the etching time determine the desired nanowire length. The NW synthesis can be optimized to vary the array length from a few hundreds of nm up to tens of microns. When above the critical percolation limit of about 60% of surface coverage, a discontinuous thin Au film realizes an interconnected network where interesting phenomena distinctive of fractals can be observed, such as light localization [28], non-linear optics effects [29], strong electrical conductance [30], and super-diffusion [31]. During the Si etching, the negative mask of the Au fractal network is molded onto the Si wafer, leading to the formation of a fractal network of nanowires. NW fractal design can be engineered with the Au deposition parameters. This top-down synthesis can be used on the wafer scale to meet the industrial productivity standard, as depicted in Figure 1d for the photo of a 4″ wafer. Moreover, the synthesis is performed at room temperature for each step for industrial compatibility. Additionally, a high surface density of about 10^11^ NWs/cm^2^ is attained. The ultrathin diameter of about 5 nm suitable for quantum confinement effects can be tuned with the metal film thickness (Figure 1e,f).

The effectiveness of quantum confinement is demonstrated with the outstanding light emission in the visible range [32]. Figure 1a shows a photo of the bright NWs’ red emission under optical pumping with a commercial blue laser pointer at 405 nm with power of 1 mW (the sample excitation is shown in the small inset on the left). Longer NW arrays obtained with higher etching time determine a brighter photoluminescence (PL), as shown in Figure 2b. We found that metal films with higher thickness define tinier uncovered Si regions, leading to smaller NW diameters with blue-shifted emission [5]. Indeed, the emission peak shifts from about 765 nm for NWs with an average diameter of about 9 nm to a wavelength of about 640 nm for NWs with 5 nm in diameter, as attested from the normalized PL intensity reported in the inset of Figure 2b. This low-cost and industrially scalable synthesis also leads to the fabrication of dense NW vertical arrays with a 2D random fractal morphology along the xy plane (Figure 2c). Fractal self-similarities and scale invariance promote electromagnetic field localization. Indeed, the lack of translational invariance leads to the spatial localization of the running waves, which are not eigenfunctions of the dilation symmetry operator. Moreover, a random fractal pattern is characterized by a self-similarity for which the structural heterogeneities are correlated on all length scales, resulting in a strong modulation of the refractive index. As a consequence, it will always be possible to match the length scale where the refractive index fluctuates for whatever effective wavelength propagates inside the medium, inducing a strong scattering extended over a broad range. In some cases, this behavior can lead to inhomogeneous localization of the electromagnetic field where both spatially localized and delocalized modes coexist. This typical property of a fractal structure results in the formation of hot-spot regions, where the intensity of the electromagnetic field is enhanced. As demonstrated for fractal antennas, broad resonances enhancing the scattering properties are attested across the self-similarity range according to the fractal geometry design. We observed in these fractal NWs an efficient light trapping extended from the visible to the near IR range due to the in-plane multiple scattering [33] and weak localization of running waves which are not eigenvectors of the dilation symmetry operator. In fact, we observed a remarkable coherent backscattering cone due to the pronounced multiple scattering, both for the elastic and Raman scattered radiation. We also demonstrated the ability to tune the system’s optical response as a function of the fractal parameters with potential for both photovoltaics and photonics applications. In fact, the strong light trapping across the fractal array determines a high enhancement for Er emission in Er:Y_2_O_3_ decorated Si NW fractal arrays obtained by glancing angle deposition. The Er light-emission intensity is enhanced as a function of the wavelength according to the fractal morphology and its heterogeneities resonances, which are responsible for the scattering of the system. This stimulating result promotes the realization of artificial fractal nanostructures with optimized optical features controlled by their geometry and design [34].

### 3.2. Label-Free and Low Cost Si Nws-Optical Sensors

Among other materials, silicon is largely diffused in the microelectronics industry and may have a tremendous impact on the commercial transfer of these technologies for biosensing. In this scenario, Si NW optical platforms emerge as easily integrable and scalable technological solutions, fostering steadier photoluminescence and mechanical stability than other confined nanostructures. The interesting structural and optical characteristics of these luminescence Si NWs synthesized by thin-film metal-assisted chemical etching has been investigated by our group for interesting application in photonics [26,27,35,36,37], and more recently for biosensing [38,39,40].

A novel class of label-free and low-cost sensors based on the luminescence of Si NWs at room temperature was reported for a plethora of targets, from proteins to pathogen genomes and exosomes, by using specific functional probes [41]. The selective capture of the desired target from the Si NW optical sensor induces the formation of non-radiative centers, quenching their luminescence proportionally to the target concentration. The luminescence quenching with respect to the sensor PL reference has been used as the sensing mechanism.

Figure 3 shows a schematic representation of the surface functionalization adopted for the native oxide surface of the pristine Si NW sensors.

We obtained three different NW optical sensors for the detection of the following:(i)small extracellular vesicles (sEV), also known as exosomes, that were selectively isolated and quantified using streptavidin and specific CD81 exosome transmembrane protein antibody as selective functionalization;(ii)the C-reactive protein (CRP) after functionalization with streptavidin and specific CRP antibody;(iii)Hepatitis B Virus (HBV) grafted with two complementary DNA probes onto GOPS-modified Si NW surface.

Recently, small extracellular vesicles as novel biomarkers able to monitor the health state of cells in human pathologies have emerged as strategic biomarkers for cancer, neurodegenerative, and cardiovascular diseases and as potential targets for therapeutic treatments or even for fertility studies [42,43,44]. However, their use for medical applications is still limited due to their isolation and the sensitivity limits of most common approaches [45], restricting their real application in liquid biopsy analysis. We realized a novel Si NWs sensor for the selective detection of sEVs, which is also able to isolate, concentrate, quantify, and analyze sEVs expressing CD81^+^ transmembrane protein from minimal volumes of biofluids, surpassing the traditional non-selective approaches. To tailor the CD81^+^ sEVs selective detection with respect to other non-specific vesicles, the following functionalization was used: 10 µg/mL of streptavidin, 50 µg/mL of biotinylated Anti-CD81, and then sample immersion in the sEV solutions at different concentrations overnight. 

We started our study by testing the effectiveness of the NW surface functionalization through confocal microscopy analyses. The laser scanning confocal microscopies for all the functionalization steps are reported in Figure 4a across the NW vertical profile. Different marking fluorophores were used for each functionalization: in yellow is reported the emission from bare Si NWs; in blue the emission of the NW profile functionalized with SA marked with Alexa 488; in red the anti-CD81 emission obtained exciting its marker fluorophore Alexa 647; in purple the sEVs CD81^+^ capture demonstration by using fluorescent lipid NBD-PE (1,2-dipalmitoyl-sn-glycero-3-phosphoethanolamine-N-(7-nitro-2-1,3-benzoxadiazol-4-yl)). The success of the functionalization is attested by perfect superposition of the emitting profiles; different excitation and detection wavelengths were used for each step to avoid emission interference.

We then tested the sensor performances with known concentrations of commercial small extracellular vesicles marked with CD81^+^ in a concentration range spanning from 10^7^ to 10^11^ sEV/mL. The room temperature PL spectrum of the bare sensors and the PL quenching at different sEV concentrations are shown in Figure 4b. For all sensors, the reference signal corresponds to the PL intensity obtained without the desired target and is strictly dependent on the surface functionalization procedure. Figure 4b shows the working principle of our NW optical sensor. As the concentration of the target species increases, the PL quenching increases proportionally, extinguishing its light emission. The dose-response curve can be fitted with Hill’s binding model [39,46], commonly used to fit the performance of the sensor based on antibody/antigen bioaffinity binding. An LOD of 2 × 10^5^ sEVs/mL was estimated from the fit, considering a concentration corresponding to two times the average error on the PL response compared to the calibration reference. The selectivity has been demonstrated by testing the Si NW sensor with non-specific synthetic vesicles (without the CD81^+^ protein on the membrane). The negative test was conducted by testing the reference sensor with nonspecific synthetic vesicles (VES) with a diameter of 100 nm. A PL variation below 4% was detected for a concentration of 3 × 10^11^ VES/mL of synthetic vesicles, thus proving the high selectivity of the sensor.

We also tested the Si NW sensor with different follicular fluid samples and with a sample of blastocoel fluid from a single embryo. This experiment demonstrated that the Si NW sensor can perfectly operate with real samples, producing results in agreement with previous measurements [47]. In this study, we confirmed that after the sEV quantification by the Si NW sensor, it is also possible to detach and selectively isolate the CD81^+^ -marked sEVs to foster their genomic analysis. The sEVs maintain their integrity after being detached, as confirmed by our measurements performed on the sEVs detached from the NW sensors, then drop-casted onto an Si support and coated with gold for the SEM analyses. In agreement with the initial morphological analyses, the detached sEVs have a typical average size of 94 ± 32 nm, as obtained from the SEM. Aside from their quantification and selective isolation, sEVs carry important genomic cargo functional for cell regulation. Indeed, RNAs represent the most promising biomarkers for analysis of sEVs, and our test verified that the PL analysis under laser pumping does not affect the RNA reading by Real-Time Polymerase Chain Reaction (RT-PCR). In fact, after isolation with the NW platform, the RNA diagnostic performed before and after the PL measurements showed no difference in miRNA concentrations in relation to the treatment with and without a laser, or for the storage protocols in Qiazol or phosphate buffer solution. These promising performances confirm that Si NW optical biosensing platforms can be used for both quantification and selective isolation of exosomes with the possibility to recover and analyze RNA and sEVs cargo, thus enabling liquid biopsy applications. 

We realized a label-free sensor based on the luminescence at room temperature of Si NWs for C-reactive protein [38,39], a primary marker for cardiovascular risk. In this case, we used a functionalization with 10 μg/mL of streptavidin and 50 μg/mL of specific biotinylated CRP antibody for CRP selective detection. This CRP sensor is capable of working over an impressive extended range of about seven orders of magnitude, from a concentration of 10^−7^ μg/mL to 0.1 μg/mL of CRP, as attested by the calibration curve reporting the integrated PL intensity as a function of the target concentration. A limit of detection (LOD) of 1.6 fM is reported, fostering the perspective for analysis in saliva (Figure 4c). The selectivity of the sensor was probed with a negative test using bovine serum albumin protein and in different biomatrices, such as human blood serum, where a high concentration of specific molecules and analytes are also present, confirming the same luminescence response (magenta star in Figure 4c). The CRP concentration in blood spans in the saturation range of the CRP sensor, above 0.01 μg/mL. We demonstrated that by doubling the functionalization concentration, the new sensor shows good flexibility of operation for higher ranges from 0.01 μg/mL up to 100 μg/mL of CRP concentration, with prospective applications for hospital analysis of blood. 

Varying the surface functional species, the NW sensor can be selective for the Hepatitis B Virus (HBV) (Figure 4d). In this case, the oxidized NW surface was modified with (3-Glycidyloxypropyl)trimethoxysilane (GOPS) with two HBV complimentary genomic bound to its final NH-amine functional groups. It is noteworthy that no PCR amplification was used for the HBV detection [40], due to the improved surface-to-volume ratio and peculiar light emission properties of quantum confined fractal Si NWs. As shown in Figure 4d, an LOD of two DNA copies of Hepatitis B Virus were attained with a PCR-free and label-free detection in a buffer environment (dark green squares). We also tested the influence of other biofluid matrices on the HBV sensor (pink rhombuses for human serum), confirming an LOD of 20 HBV copies for clone DNA in a human serum environment. The selectivity effectiveness of the HBV NW sensor was tested in a high concentration of 2000 copies of non-specific mycobacteria tuberculosis (MBT) DNA both in buffer (magenta square) and in human serum (red rhombus), and no significant shift from the reference signal was observed.

Finally, Figure 4e shows the realization of square wells of Si NWs for the manufacture of arrays of optical biosensors able to detect on a single platform a variety of targets simply varying the functionalization by spotting in the desired well. Our idea is to realize a comprehensive Si NW biosensor for the simultaneous screening of different targets with the possibility of realizing an easy optical reader for diagnosis. The characteristics of Si NWs optical biosensors are reviewed and compared in Table 1.

Standard immunoassay methods for protein analysis are Western blot or ELISA, the first one characterized by a limit of detection of about 0.3 mg/mL for standard approaches [48], and the latter with an LOD of 1 ng/mL of protein analyte (or 10^−11^ M, protein mass: 100,000 Dalton), and antibody–antigen binding constant K of 10^8^ M^−1^ [49]. However, nowadays, several applications require improved detection limits down to the pico and femtomolar range for novel noninvasive assays. For the pathogen genome, the standard detection method is the real time polymerase chain reaction, proving end-point analysis with limits of detection of approximately 30 copies of the genome per 1 μL or purified DNA solution after about 35 cycles of amplification [50]. Despite its sensitivity, PCR is expensive, requiring specialized personnel and laboratories that are not easily portable. Hence, novel sensing platforms able to push further the protein or genome limit of detection that can be commercially transferred in low-cost point-of-care devices are needed. 

The surface functionalization, detection ranges, and the LODs and their comparison with the literature are reported in Table 1 for all the presented applications. The sensitivity of a sensor is established as the relationship between the change in analyte concentration and the intensity of the signal generated from the transducer. For all of our NW biosensors, we measured a sensitivity of about 10% in the linear range. In particular, for CD81-sEV recognition, a sensitivity of about 10% is attained in the linear range from 10^8^ to 10^10^ sEV/mL of the target concentration. The CRP-biosensor shows a sensitivity of about 10% in the linear range from 10^−7^ to 10^−3^ μg/mL, and the HBV-biosensor has a 10% sensitivity from 2 to 10^−3^ cps.

Thus, the actual sensor market opportunities are the realization of platforms with outstanding performance and novel platforms pushing the sensing applications outside hospitals with portable and point-of-care devices. Silicon nanostructured sensors combine the strategic advantage of novel physical phenomena arising from quantum confinement to the increment of the exposed surface-to-volume ratio. Despite several issues that still need to be addressed, Si nanostructures pave the way toward smarter, more interconnected, and portable platforms in the exciting field of sensing devices. Many transduction mechanics are promising for Si nanostructured biosensors, starting from electronic transduction with Si nanowires field-effect transistors and porous Si, to optical ones based on porous silicon, photonics crystals, luminescent Si quantum dots, and luminescent Si NWs. Among others, light-emitting Si nanowire sensors demonstrated remarkable fM performance with an easy optical readout, making them a natural candidate for novel biosensors. Most importantly, Si NW fabrication is advancing rapidly in performance and capability, meeting the standard of compatibility with complementary metal-oxide semiconductor foundry processes, and merging low-cost and scalable fabrication for optical device miniaturization. Therefore, the realization of high precision optical sensing solutions fostering small size, mass production availability at a low cost, robustness, stability, and possibility for lab-on-chip applications is extremely appealing compared to the other Si nanostructures. It is not easy to forecast which Si NW-based platforms will hit the market in the near future and it depends on the applications.

## 4. Conclusions

In this paper, we demonstrated the prospective applications of quantum confined fractal Si nanowires for Si-based photonics due to their impressive optical response. Light-emitting ultrathin Si NWs obtained by our thin metal film fabrication meet the standard for industrial implementation, fostering the manufacture of exotic fractals where multiple scattering regime and weak localization of light are observed. Novel artificial 2D fractal arrays decorated with optically active Er:Y_2_O_3_ demonstrate the relationship between Er enhanced optical responses and the fractal morphology and scattering for applications in the field of low-cost disordered photonic and multiplexed sensing. This result opens the way to the realization of original and controllable artificial fractals that integrate interesting elements in a silicon microelectronics-compatible substrate. In conclusion, these Si NW optical platforms find appealing employment as label-free and PCR-free sensors for the selective detection of protein and pathogen genomes, with an extremely low limit of detections of fM and a few copies, respectively. Additionally, the selective quantification and isolation of small extracellular vesicles, such as CD81^+^ exosomes, is also reported, surpassing the traditional approaches for the isolation, concentration, and quantification of vesicles. Indeed, a limit of detection of about 2 × 10^5^ Ex/mL was fitted for this Si NW sensor. Additionally, we demonstrated the possibility to measure even very small volume samples as the blastocoel fluid. Moreover, their genomic screening for RNA diagnostic fosters the quantification and genomic analysis of a selected class of extracellular vesicles, also opening manifold future applications for liquid biopsy. The proper development of these optical Si NW biosensing platforms would allow the production of an industrial device of great impact for the patient for the early diagnosis of the pathology at home, which can be sent to hospital facilities only if needed. Moreover, the easy readability of optical Si NW biosensing platforms and their low cost would allow for early screening of diseases in those third-world countries where hospitals and specialized facilities are not easily accessible to the population.

## Figures and Tables

**Figure 1 biosensors-12-01052-f001:**
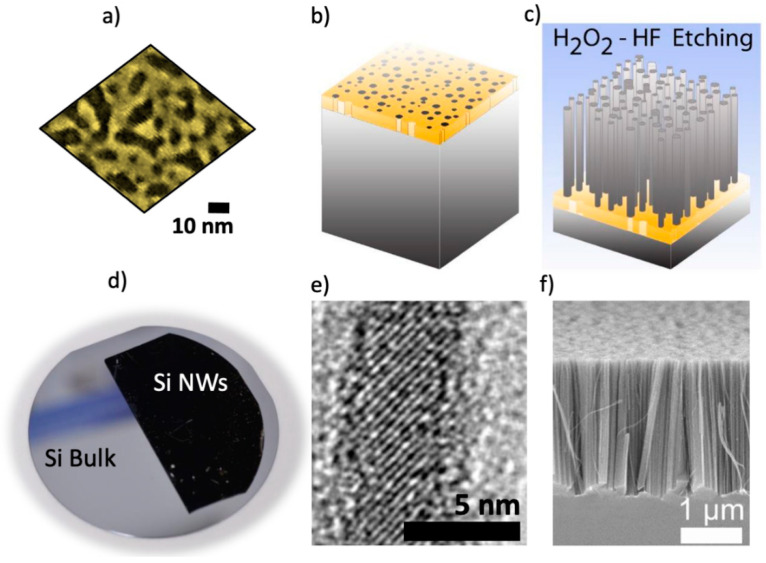
(**a**) 2 nm-thick Au percolative film deposited by electron beam evaporation at room temperature onto Si wafer. (**b**,**c**) Schematic representation of Au deposition and Au/Si wet etching. (**d**) Photo of a 4” Si wafer whose black half is composed of Si NW array. (**e**) TEM microscopy attesting the quantum confined diameter and (**f**) SEM Cross-section image of the 3 µm long NW vertical array.

**Figure 2 biosensors-12-01052-f002:**
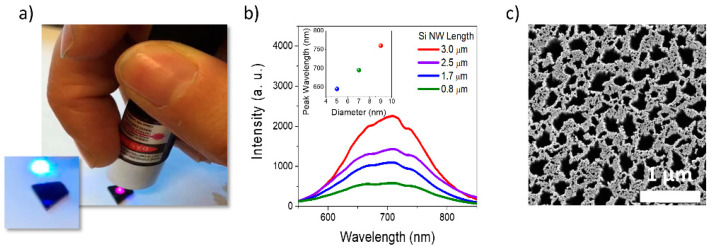
(**a**) Photo attesting the bright room temperature emission observed by the naked eye and under excitation by a commercial laser pointed at a wavelength of 405 nm, the inset shows the same laser onto white paper. (**b**) Photoluminescence emission of NWs with the same diameter but different lengths, while the inset shows the normalized photoluminescence emission from Si NWs with a different average diameter under excitation with 488 nm line from an Ar^+^ laser. (**c**) SEM plan-view of 2 µm long fractal Si NW array showing the fractal design along the xy plane.

**Figure 3 biosensors-12-01052-f003:**
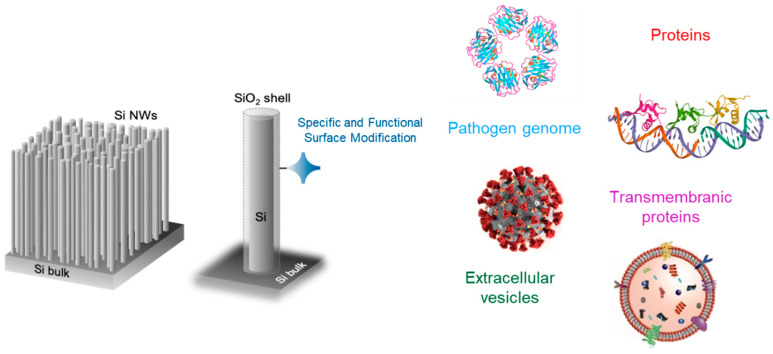
Schematic representation of the selective functionalization adopted for the Si NW optical sensors to detect a variety of different targets.

**Figure 4 biosensors-12-01052-f004:**
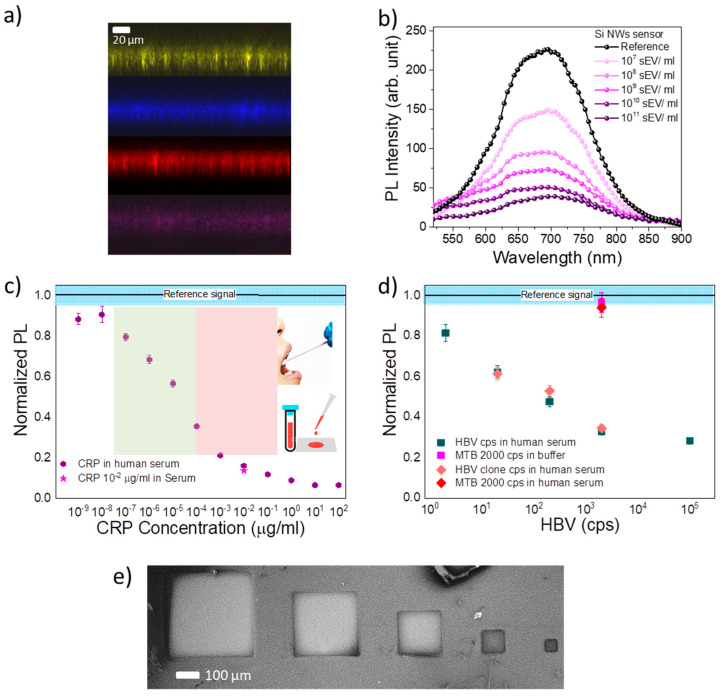
(**a**) Confocal microscopies of all the functionalization steps for sEV detection by using different fluorophores: bare Si NW with yellow emission, SA emission in blue obtained exciting its Alexa488 marker, anti-CD81 emission in red obtained exciting its marker fluorophore Alexa 647, sEVs CD81^+^ capture by using NBD-PE-marked sEVs whose emission is shown in purple. (**b**) Photoluminescence quenching of the NW sensor for different sEV concentrations. (**c**) Calibration curve of NW sensor for C-reactive protein detection. (**d**) Calibration curve of NW sensor for Hepatitis B Virus. (**e**) SEM plan-view microscopy of squared wells for the Si NWs local synthesis for optical biosensor arrays realized by 365 nm optical lithography process in hard contact mode with a Karl Suss MA6 tool.

**Table 1 biosensors-12-01052-t001:** Characteristics of Si NW optical biosensors.

Si NW Optical Biosensors for	Surface Functionalization	Detection Range	LOD	Standard Methods LOD
C-reactive Protein	10 μg/mL SA + 50 μg/mL CRP-Ab for saliva screening	10^−8^–1 μg/mL	10^−7^ μg/mL (1.6 fM)	0.3 mg/mL Western Blot [48]
	20 μg/mL SA + 100 μg/mL CRP-Ab for serum screening	10^−2^–100 μg/mL	10^−1^ μg/mL (nM)	10^−1^ ng/mL ELISA [49]
Hepatitis B Virus	GOPS + HBV complimentary probes	2–10^5^ cps	2 cps in buffer 20 cps in serum	30 cps per 35 cycles RT-PCR [50]
Small Extracellular Vesicles	10 μg/mL SA + 50 μg/mL CD81-Ab	10^5^–10^11^ sEVs/ml	10^5^ sEVs/ml	10^8^ sEVs/mL ELISA nPLEX [51]

## Data Availability

Data is contained within the article.

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
