# Peer review of "Future Prospects of Luminescent Silicon Nanowires Biosensors"

_biosensors, 2022, doi:10.3390/bios12111052_

Round 1

Reviewer 1 Report

In this work, the authors demonstrated the perspective applications of quantum confined fractal Si nanowires for Si-based photonics due to their impressive optical response. Si NW optical platforms appealing employment as label-free and PCR-free sensors for the selective detection of protein and pathogen genome because of extremely low limit of detections of fM. Contents are interesting, and I can recommend a publication in present form.

Author Response

Reviewer 1

In this work, the authors demonstrated the perspective applications of quantum confined fractal Si nanowires for Si-based photonics due to their impressive optical response. Si NW optical platforms appealing employment as label-free and PCR-free sensors for the selective detection of protein and pathogen genome because of extremely low limit of detections of fM. Contents are interesting, and I can recommend a publication in present form.

We thank the referee for his/her positive consideration of the manuscript.

Reviewer 2 Report

Comment for authors

The manuscript “Future perspective of luminescent Si nanowires-based biosensing devices” seem interesting but needs modification before acceptance.

*Title needs to be modified, it's out of the format.

*Abstract. The author has to be concise in a way for better insight into the problem along with objectives and output.

*There are some typos and grammatical errors that need to be addressed properly. Present the consistency. Authors are strongly suggested to seek professional help. The expression of many sentences is too long to understand in a manuscript. Concise them into smaller ones.

*How does the author compare the novelty and comparison statement from the previously published review paper?

*There should be a comparison with the standard method.

*How about the practical application and impact of research?

*Figure 2B needs to be smooth.

Author Response

Reviewer 1

In this work, the authors demonstrated the perspective applications of quantum confined fractal Si nanowires for Si-based photonics due to their impressive optical response. Si NW optical platforms appealing employment as label-free and PCR-free sensors for the selective detection of protein and pathogen genome because of extremely low limit of detections of fM. Contents are interesting, and I can recommend a publication in present form.

We thank the referee for his/her positive consideration of the manuscript.

Reviewer 2

The manuscript “Future perspective of luminescent Si nanowires-based biosensing devices” seem interesting but needs modification before acceptance.

We thank the referee for his/her careful reading of the manuscript, we agree with the proposed improvement and modified the revised version of the manuscript accordingly.

*Title needs to be modified, it's out of the format.

According to the journal regulation: “the title of your manuscript should be concise, specific and relevant. It should identify if the study reports (human or animal) trial data, or is a systematic review, meta-analysis or replication study. When gene or protein names are included, the abbreviated name rather than full name should be used. Please do not include abbreviated or short forms of the title, such as a running title or head”

Hence, we modified the title of the revised paper from “Future perspective of luminescent Si nanowires-based biosensing devices” to “Future perspective of luminescent silicon nanowires biosensors” according to the referee’s suggestions.

*Abstract. The author has to be concise in a way for better insight into the problem along with objectives and output.

We thank the referee as we feel that this suggestion improves conveying the right message to the reader. We modified the abstract in the revised paper based on the referee’s instructions as follows:

“In this paper, we exploit the perspective of luminescent Si nanowires (NWs) in the uprising field of commercial nanodevices for biosensing for the selective recognition of proteins and pathogen genome. We fabricated quantum confined fractal arrays of Si NWs with room temperature emission at 700 nm obtained by thin-film metal-assisted chemical etching with high production output at low cost. The fascinating optical features arising from multiple scattering and weak localization of light promote the use of Si NWs as optical biosensing platforms with high sensi-tivity and selectivity. In this work, label-free Si NWs optical sensors are surface modified for the selective detection of C-reactive protein through antigen/gene interaction. In this case, we report the lowest limit of detection (LOD) of 1.6 fM, fostering the flexibility of different dynamic ranges for detection either in saliva or for serum analyses. By varying the NW surface functionalization with the specific antigen, the luminescence quenching of NW biosensors is used to measure the hepatitis B-virus pathogen genome without PCR-amplification, with LOD of about 20 copies in real samples or blood matrix. The promising results foster NW optical biosensors to detect and isolate extracellular vesicles (EV) marked with CD81 protein with unprecedented sensitivity (LOD 2×105 sEV/ml) enabling their measurement even in a small amount of blastocoel fluids.”

 *There are some typos and grammatical errors that need to be addressed properly. Present the consistency. Authors are strongly suggested to seek professional help. The expression of many sentences is too long to understand in a manuscript. Concise them into smaller ones.

We appreciate the referee’s suggestion, and we extensively edited the revised manuscript correcting the typo and grammar errors throughout the whole paper accordingly. The paper was also improved in terms of readability by simplifying long sentences into shorter ones.

*How does the author compare the novelty and comparison statement from the previously published review paper?

We thank the referee for his/her constructive criticism. We can find in the literature very comprehensive reviews of Si-based nanostructured sensors. However, other reviews commonly discuss Si NW sensor only for what concerns an electrical (such as in a Field Effect Transistor configuration) or electrochemical transduction. Instead, this review aims to discuss, the field of Si NW biosensors involving a fluorescent transduction mechanism. Indeed, in the literature, there are only a few reports on luminescent silicon nanowires employing expensive equipment not implantable with the industrial scale. There are other works on luminescent nanowires obtained with other materials that again are not comparable with the Si industrial technology, its low-cost availability, and commercial processes.

Hence, we added the following sentences on page 1 and page 2 of the revised version of the manuscript:

“Among different transduction methods, luminescent sensors emerge as some of the outperforming approaches due to their easy data interpretation, affordability, and real-time analyses.”

“Common luminescent NWs sensors are in general based on non-emitting Si NWs used as passive substrates and coupled with luminescent probes. While more interesting approaches employ NWs as quenchers in stem-loop configuration and have recently been used for direct luminescence sensing. In this framework, Si NWs with improved compatibility with Si standard technology and flat architecture fabrication arises as a promising solution for low-cost, scalable, and non-toxic, biosensing devices. Indeed, our Si NWs couple their unique physical and optical properties of room temperature luminescence arising from quantum confinement with the high surface-to-volume ratio of about 103 combined with an easy commercial transfer.”

*There should be a comparison with the standard method.

As suggested by the referee, we compared our sensing performance with other standard methods for proteins, DNA, and extracellular vesicle detection, such as Western-Blot, PCR, or ELISA.

Hence, we added the following sentences on page XX of the revised version of the manuscript:

“Standard immunoassay methods for protein analysis are western blot or ELISA, the first one characterized by a limit of detection of about 0.3 mg/ml for standard ap-proaches [Koch, R.J. et al., Validating Antibodies for Quantitative Western Blot Measurements with Microwestern Array. Scientific Reports 2018 8:1 2018, 8, 1–10, doi:10.1038/s41598-018-29436-0.], and the latter with LOD of 1 ng/ml of protein analyte (or 10−11 M, protein mass: 100 000 Dalton), and antibody–antigen binding constant K of 108 M−1 [Zhang, S. et al., Predicting Detection Limits of Enzyme-Linked Immunosorbent Assay (ELISA) and Bioanalytical Techniques in General. Analyst 2013, 139, 439–445, doi:10.1039/C3AN01835K]. How-ever, nowadays several applications require improved detection limits down to the pico and femtomolar range for novel noninvasive assays. For pathogen genome, the standard detection method is the real time polymerase chain reaction, proving end-point analysis with limits of detection of approximately 30 copies of the genome per 1 μl or purified DNA solution after about 35 cycles of amplification [Purcell, R. V. et al., Comparison of Standard, Quantitative and Digital PCR in the Detection of Enterotoxigenic Bacteroides Fragilis. Scientific Reports 2016 6:1 2016, 6, 1–8, doi:10.1038/srep34554]. Despite its sensitivity, PCR is expensive, requiring specialized personnel and laboratories that are hardly portable. Hence, novel sensing platforms able to push further the protein or genome limit of detection that can be commercially transferred in low-cost point-of-care devices are needed.”

*How about the practical application and impact of research?

We thank the referee as we feel that our paper was lacking in this aspect. In this paper, we wanted to demonstrate the proof of concept of this sensing technology. Indeed, our low-cost luminescent silicon nanowires meet the requirements for Si standard integration and scalable industrial production.

With the proper development of these optical Si NWs biosensing platforms, it would be possible to produce an industrial device of great impact for the patient, as he could perform the early diagnosis of the pathology at home in self-autonomy, recurring to hospital facilities only if needed. Moreover, the easy readability of optical Si NWs biosensing platforms and their low cost would allow for early screening of diseases in those third-world countries where hospitals and specialized facilities are not easily accessible to the population.

Hence, we added the following sentences on page 10 of the revised version of the manuscript:

“The proper development of these optical Si NWs biosensing platforms would allow producing an industrial device of great impact for the patient for the early diagnosis of the pathology at home in self-autonomy, recurring to hospital facilities only if needed. Moreover, the easy readability of optical Si NWs biosensing platforms and their low cost would allow for early screening of diseases in those third-world countries where hospitals and specialized facilities are not easily accessible to the population.”

*Figure 2B needs to be smooth.

In accordance with the referee suggestion, we performed a light smoothing of our raw photoluminescence data averaging with Savitzky–Golay on 250 points, and preserving the original information of the luminescent spectra.

Reviewer 3

The manuscript presents label-free Si nanowires optical sensors to detect some targets. The results revealed that the detection limit was obtained about 2×105 Ex/ml for the proposed Si nanowire biosensor. The title of manuscript is fair and the structure of biosensor is well presented, however, the manuscript needs to be revised considering the following comments:

  1. The manuscript should be carefully edited. For example Figure 3 is described in Line 289, after Figure 4. Also, there are some typo/grammatical errors which should be corrected.

We thank the referee for his/her careful reading of the manuscript. We extensively edited the revised manuscript correcting the typo and grammar errors throughout the whole paper. We also corrected the Figure citation in the right order as suggested by the referee.

  1. In Lines 49-52: "In this framework, Silicon nanowires (NWs) are considered one of the most outstanding building blocks for emerging nanodevices, finding flexible applications in electronics [6,7], energy [8–10], photonics [11,12], and sensing [13–15]." It is recommended to add other important photonic structures for solar cells such as 10.5277/oa180409; 10.1016/j.jmat.2018.11.007.

The references suggested are indeed of great interest, and were added to the revised version of the manuscript on page 2 as references [11] and [14] :

  1. Sahoo, M.K.; Kale, P. Integration of Silicon Nanowires in Solar Cell Structure for Efficiency Enhancement: A Review. Journal of Materiomics 2019, 5, 34–48, doi:10.1016/J.JMAT.2018.11.007.

  1. Olyaee, S.; Farhadipour, F. Investigation of Hybrid Ge QDs/Si Nanowires Solar Cell with Improvement in Cell Efficiency. Optica Applicata 2018, XLVIII, doi:10.5277/oa180409.
  2. Please clarify selecting Si nanowire for this structure briefly (InP may be better material for this structure?). 

We focus our attention on luminescent Si Nanowires as silicon is the leading material for the electronic industry, it is easily available at low fabrication cost and unlike others. Moreover, the great advantage it is not toxic, while other materials like indium phosphide or cadmium selenide, not only are really expensive but also toxic, posing a threat to the patient health.

We feel that this aspect was not very clear in the paper, hence, according to the good suggestion of the referee we added the following phrase in the revised version of the manuscript on page 1:

“Compared to other materials, Si availability and performance offer a plethora of industrial technologies for modern device realization. Indeed, being easily available at low fabrication cost, silicon is the leading material for the electronic industry and commercial devices. Moreover, it has the great advantage of not being toxic, unlike other materials such as indium phosphide or cadmium selenide, which are also really expensive and toxic, posing a threat to the patient health.”

  1. All symbols and abbreviations should be defined, for example PBS, HPLC, etc.

The referee is absolutely right, we checked the whole text and amended the missing abbreviations in the revised manuscript.

  1. The detection limit is reported as a key parameter of the sensor, but other parameters are missed. The sensitivity, quality factor, and range of detection are other parameters that need to be explained and addressed.

We thank the referee for the careful reading and indeed, he/she is right. We remark that the quality factor is a parameter used for other photonic devices as photonic crystals, and it is not applicable in our case. Moreover, our synthesis demonstrates a high reproducibility and uniformity for the NWs fabrication. Each sensor calibration curve was obtained after a proper statistic of more than 5 measures onto the same sample and more than 3 samples for each target concentration. Moreover, the results are obtained using the same Si NW batch with the calibration realized normalizing the PL quenching to the functionalized and not exposed to the target sample. All these considerations were used to increase the affordability of the sensors.  

We revised the manuscript implementing the missing information on the sensitivity in terms of PL quenching percentage variation with respect to the error in the text. In literature, the sensitivity is scarcely reported, however, we estimated a sensitivity variation of about 10% PL quenching for each order of magnitude or the target concentration. By considering this point we also reported the Signal/Noise ratio as the PL quenching variation compared to the standard deviation obtained per concentration.  

The detection ranges and the sensor selectivity are commented in the text, and we added the missing information of the detection ranges and the standards method LODs in table 1 on page 10 for all the presented applications comparing them to the literature as requested.

We added the missing information on the sensitivities of the sensors adding the following sentence on page 10:

“The surface functionalization, detection ranges, the LODs and their comparison with the literature are reported in table 1 for all the presented applications. The sensitivity of a sensor is established as the relationship between the change in analyte concentration and the intensity of the signal generated from the transducer. For all of our NW biosensors, we measured a sensitivity of about 10% in the linear range. In particular, for CD81-sEV recognition, a sensitivity of about 10% is attained in the linear range from 108 to 1010 sEV/mL of the target concentration. The CRP-biosensor shows a sensitivity of about 10% in the linear range from 10−7 to 10−3 μg/mL, and the HBV-biosensor has a 10% sensitivity from 2 to 10−3 cps.”

  1. The results compared in Table 1 should be compared with other reports. Please also explain how these results were verified.

The referee is right and a comparison with other detection methods was lacking. So, we added a column for comparison to the standard detection limits to table 1 in the revised manuscript.

We added the following sentence on page 3 of the new manuscript to explain how the results were verified as requested:

“All photoluminescence (PL) measurements were performed by focusing the 364 nm or line of an Ar+ laser onto the sample through a 60×UV Olympus objective at a laser power of about 80 μW measured on the sample. PL spectra were acquired by a HR800 spectrometer (Horiba Jobin Yvon) coupled to a cooled CCD detector. For each concentration, PL measurements were averaged over 5 different points of the same device and evaluated also on three sample replicates of different Si NW platforms. The calibration curves have been obtained by repeating the PL measurements several times on different sensors. The relative errors over the replicates are lower than 5%.”

Reviewer 3 Report

The manuscript presents label-free Si nanowires optical sensors to detect some targets. The results revealed that the detection limit was obtained about 2×105 Ex/ml for the proposed Si nanowire biosensor. The title of manuscript is fair and the structure of biosensor is well presented, however, the manuscript needs to be revised considering the following comments:

1. The manuscript should be carefully edited. For example Figure 3 is described in Line 289, after Figure 4. Also, there are some typo/grammatical errors which should be corrected.

2. In Lines 49-52: "In this framework, Silicon nanowires (NWs) are considered one of the most outstanding building blocks for emerging nanodevices, finding flexible applications in electronics [6,7], energy [8–10], photonics [11,12], and sensing [13–15]." It is recommended to add other important photonic structures for solar cells such as 10.5277/oa180409; 10.1016/j.jmat.2018.11.007.

3. Please clarify selecting Si nanowire for this structure briefly (InP may be better material for this structure?). 

4. All symbols and abbreviations should be defined, for example PBS, HPLC, etc.

5. The detection limit is reported as a key parameter of the sensor, but other parameters are missed. The sensitivity, quality factor, and range of detection are other parameters which need to be explained and addressed.

6. The results compared in Table 1 should be compared with other reports. Please also explain how these results were verified.

Author Response

(The authors gave the same response as above.)

Round 2

Reviewer 3 Report

The manuscript is revised according to the comments, and now, it can be accepted for publication in Biosensors. Congratulation!